# Effect of Pen Space Allowances on Growth Performance of Finishing Pigs [note 1]

**DOI:** 10.3390/ani15101451

**Published:** 2025-05-17

**Authors:** Ryan S. Samuel, Joseph E. Darrington, Benoit St-Pierre, Crystal L. Levesque, Robert C. Thaler

**Affiliations:** Department of Animal Science, South Dakota State University, Brookings, SD 57007, USAbenoit.st-pierre@sdstate.edu (B.S.-P.); crystal.levesque@sdstate.edu (C.L.L.); robert.thaler@sdstate.edu (R.C.T.)

**Keywords:** housing, swine, production, floor space, market weight, carcass, finishing pigs

## Abstract

This study investigated how different amounts of floor space per pig affect their growth and carcass traits, given that current guidelines are based on lighter pigs. Three space allocations were tested: 0.61, 0.75, and 0.88 square meters per pig. The results show that overall feed intake, body weight, and feed efficiency were not significantly affected by pen space. However, pigs given less space (0.61 m^2^) tended to weigh less at the barn average of 135 kg but caught up in weight by final marketing. Pigs with less space also had slightly higher carcass lean percentage.

## 1. Introduction

The number of finishing pigs per pen should maximize floor space utilization but must consider the importance of pen space allocation on animal welfare [1], as well as the production efficiency of growing pigs in terms of income over feed and facility costs [2] or net revenue [3]. Animal performance and welfare (e.g., pen hygiene, maintenance of health, and ability to express natural behaviors) generally decline as the space allocation per pig goes below a lower critical value [1,4,5]. Johnston et al. [6] observed that floor space restriction caused negative effects on animal performance close to marketing. Furthermore, as pigs were marketed at heavier weights, greater floor space per pig was necessary to mitigate any negative effects. 

An extensive literature review reported that pig housing environments with limited space availability cause pen scale reductions in ADFI, which correspond with reduced ADG, primarily due to competition for feeder and water access [7]. The lower critical value of space allocation below which animal performance declines was defined at k = 0.0336, where k is the fraction of total body surface area allocated as floor space area per pig [4]. For finisher pigs weighing 127 kg (the current average market weight for commercial hog operations in the USA), this corresponds to a calculated minimum space allocation of 0.86 m^2^/pig to avoid adverse effects on production. Buhr et al. [8] reported an industry average of 0.67 m^2^/pig at market weight, which is within previous space/pig recommendations [9,10,11]; however, these recommendations are almost 25 years old and based on considerably lighter market weight targets. Thus, at current production practices, it is likely that significant losses in production efficiency may occur in the finisher phase. Compounding the concern with potential losses is a trend for increasing target market weights (i.e., >135 kg) [6]. The drive for heavier market weights is economically based and represents a method of distributing fixed costs of the sow herd by increasing the total pounds of pork produced per sow per year [2,3,7]. In an economic model developed to optimize market weight strategies, Buhr et al. [8] concluded that the most effective strategy to reduce the negative impacts of space limitation was to increase the available space through lower stocking densities or additional physical floor space in the form of more barns. These strategies are impractical in the short term and assume a clear understanding of the optimal k-value at heavy pig weights (i.e., >135 kg) and the economic and production efficiency implications of space limitation on heavy-weight pigs. Precise information regarding economically optimal space allocations to produce heavy finisher hogs and incorporation of enrichment opportunities will guide the design of tomorrow’s confinement buildings [1].

The objective of this study was to investigate the impact of pen space allowances on the growth performance of finishing pigs raised to over 135 kg. 

## 2. Materials and Methods

### 2.1. Animal Management and Diets

The experimental procedures for this study were reviewed and approved by the Institutional Animal Care and Use Committee of South Dakota State University (SDSU, IACUC #2005-026E). Animals were housed at the SDSU Offsite Wean-to-Finish barn, which is a mechanically ventilated facility housing approximately 1200 pigs in pens of equal size (3.1 × 6.9 m). Pigs (*n* = 1035; initial body weight (BW) 108 ± 0.7 kg) were the offspring of Babcock Genetics maternal and paternal lines and were allocated into 45 pens of 23 pigs per pen according to sex such that there were 15 pens each of gilts, barrows, and mixed sex such that floor space and not pen group size was the dependent variable tested. The mixed-sex pens were approximately equal in gilts and barrows at the start of the trial. Feed and water were provided ad libitum from a five-slot dry feeder (178 cm total length; SD Industries, Alexandria, SD, USA) with feed delivered by a single M-Series FEEDPro system (FeedLogic by ComDel Innovation, Wahpeton, ND, USA) and water available via two cup waterers per pen, respectively.

Consistent with commercial-scale standard operating procedures, feed and water availability, as well as the animals, were monitored daily. Any veterinary treatments were recorded, including the reason for treatment, the number of pigs treated per pen, the administered drug dosage, and duration of treatment, as well as the reason for removal of any animals, including death or due to an untreatable health condition such as umbilical hernia or injury.

Diets consisting of corn and soybean meal as the major feedstuffs were fed in two phases within the trial: (1) 100 to 120 kg providing 0.57% digestible lysine and (2) 120 to 140 kg providing 0.48% digestible lysine (Table 1). The diets were formulated to provide nutrients to meet or exceed the recommendations of NRC [12] based on expected rate of gain, feed intake, and BW.

### 2.2. Pen Space Allocations

Approximately one month before the start of the pen space study described in this report, pigs from the barn were reallocated to create 15 barrow, 15 gilt, and 15 mixed-sex pens, with the mixed-sex pens housing approximately equal numbers of gilts and barrows. Barrow, gilt, and mixed pens were randomly assigned within the facility. At the start of and for the duration of the 49-day trial, pen space was adjusted by changing the position of the front gates. Pens with the front gate adjusted as far back as the feeder provided 0.61 m^2^/pig, while pens with the front gate placed equidistant between the standard position and the furthest back position provided 0.75 m^2^/pig; the front gate of the pen in the standard position provided a pen space of 0.88 m^2^/pig. Each space allocation was replicated 5 times within each sex, resulting in a total of 15 pens per space allocation. 

### 2.3. Measurements and Calculations

Total BW of the pigs within each of the pens was measured weekly using a scale (Digi-Star EZ400, Ft. Atkinson, WI, USA) that accommodated the entire pen at once. The average BW for the pen was calculated as$$\mathrm{BW}=\frac{\;\mathrm{Total}\;\mathrm{pen}\;\mathrm{weight}}{\;\mathrm{Number}\;\mathrm{of}\;\mathrm{pigs}\;\mathrm{in}\;\mathrm{pen}}$$

From the differences in BW, average daily gain (ADG) for each pen was calculated.$$\mathrm{ADG}=\frac{{\mathrm{BW}}_{2}{-\mathrm{BW}}_{1}}{\;\mathrm{days}}$$

The volume of feed remaining in the feeder was estimated according to a previously determined equation:FL = 0.3863 × X^2^ + 3.5509 × X + 22.849
where FL = feed leftover in the feeder, and X = the measurement of empty space in the feeders (inches). From the feed delivery and feed remaining information, the average daily feed intake (ADFI) was calculated for each pen.

### 2.4. Marketing and Carcass Measurements

Marketing of individual animals began after the average BW in the barn exceeded 135 kg. The heaviest animals in each pen were identified by weighing pigs individually one or two days before the pigs were scheduled to be transported. Individual animals were marked on their back with one of three colors to designate the space allocation treatment (i.e., 0.61, 0.75, or 0.88 m^2^/per pig) that had been applied before loading onto standard pot-belly animal transport trailers over three weeks. Six, seven or eight, and five pigs per pen were removed six and six days apart. The hot carcass weight (HCW) and carcass lean percent were reported by the commercial abattoir according to the color group from 330, 282, and 332 pigs housed with 0.61, 0.75, or 0.88 m^2^/per pig, respectively.

### 2.5. Statistical Evaluation

Data are presented in tables as least square means ± standard error of the mean (SEM), unless otherwise stated. A normal distribution of the data was tested using the UNIVARIATE procedure, and an appropriate statistical analysis of variance allowing for both fixed and random effects fitting a mixed linear model was performed using PROC MIXED (SAS Inst. Inc., Cary, NC, USA). The influence of sex on BW was investigated considering ‘sex’ as the fixed effect and ‘pen’ as a random variable in the model. Space allocation was investigated as a fixed effect in the model to evaluate the influence on weekly ADG, ADFI, HCW. As above, to account for variability between pens on subsequent weigh days without attributing it to time-dependent factors, the experimental unit ‘pen’ was included as a random variable in the model. Individual animals were considered the experimental unit in the analysis of HCW. Model statements were tested using the Kenward–Roger degrees of freedom method. Least square means were compared using the ‘pdiff’ option. Significance was taken at *p* < 0.05, while 0.05 ≤ *p* < 0.1 was regarded as a tendency. 

## 3. Results

### 3.1. Animal Performance

#### 3.1.1. Body Weight by Sex

Pigs housed in barrow-only pens had greater (*p* < 0.01) average BWs compared with pigs housed in mixed-sex or gilt-only pens at the initiation of the trial and up to marketing of the heaviest animals began (Table 2). In fact, the majority of the first pigs removed from the barn in the first marketing event were barrows. All pigs were weighed up to d37 before pigs were started to be shipped for marketing. Thereafter, no differences in BW according to sex were observed.

#### 3.1.2. Body Weight by Pen Space Allocation

There were no differences (*p* > 0.34) in pig BW when finishing pigs were housed in pens providing 0.61, 0.75, or 0.88 m^2^ per pig until the average BW was greater than 135 kg (Table 3). Thereafter, the average BW of pigs allocated 0.61 m^2^ tended (*p* ≤ 0.09) to be lower than the average BW of pigs in the other pen space allocations until the final weigh day, when BW was again not different according to pen space.

#### 3.1.3. Average Daily Gain by Pen Space Allocation

The ADG was reduced (*p* < 0.01) when pigs were provided the lowest pen space allocation during the third week of the trial (Table 4). Previously and thereafter, there was no effect of pen space allocation on ADG.

#### 3.1.4. Feed Intake by Pen Space Allocation

The ADFI was greater (*p* = 0.03) for pigs provided 0.75 m^2^ compared with pigs provided 0.61 m^2^, and the ADFI of pigs provided 0.88 m^2^ was intermediate during the third week of the trial, just before marketing of the heaviest five animals per pen began (Table 5). 

#### 3.1.5. Gain:Feed

Pigs allocated 0.61 m^2^ tended (*p* ≤ 0.08) to have reduced gain:feed compared with pigs allocated 0.88 m^2^ during the first two weeks of the trial (Table 6).

#### 3.1.6. Hot Carcass Weight, Carcass Lean Percent, and Carcass Value by Pen Space Allocation

The overall HCW was not different between treatments (Table 7). The measured carcass lean percent tended (*p* = 0.08) to be greater (56.8 vs. 56.5%) from pigs provided 0.61 m^2^/pig of floor space compared with pigs provided 0.88 m^2^/pig, thus improving (*p* = 0.03) the carcass value (USD 60.52 vs. USD 59.71/45 kg) of those animals.

## 4. Discussion

The study objective was to investigate pen space allowances on the growth performance of finishing pigs raised to over 135 kg. The BW of finishing pigs was not impacted by the space allocations tested in this trial until pigs weighed more than 135 kg. Thereafter, once the heaviest pigs had been removed according to the marketing plan, there were no differences in BW. Similarly, barrow only pens weighed more from the beginning of the trial up to the first marketing, as previously reported [13]. Although ADG was lower for pigs housed in the lowest floor space allocation in the third week of the trial, before and after that point, there were no observed differences between space allocations. Interestingly, the ADG and ADFI of pigs the week after marketing, regardless of pen space allocation, were numerically increased compared with other weeks. The reduction in space likely made it more difficult for pigs to access feed and water freely, which could negatively affect growth before they reached their heaviest weights [6]. Consistent with BW, ADFI tended to be lower for pigs housed in pens with the smallest space allowance compared with pigs housed in pens with the largest space allowance the week before marketing, with no other differences. Therefore, it appears that the standard marketing procedure of removing the heaviest pigs from each pen before the average BW reaches 135 kg is sufficient to eliminate any negative influence of pen space allowance on the growth performance of finishing pigs.

Another concern with heavy market weights is the further increase in weight variability at the barn and pen level, resulting in potential penalties from the packer for either light- or heavy-weight pigs when marketed. Buhr et al. [8] assumed intra-pen variability was constant and used a coefficient of variation of 9.5%. The value 9.5% was based on historic University of Nebraska—Lincoln research barn data, where pigs were marketed in a single day or week (i.e., ‘dumped’). Not many swine operations market their finishers as dump hogs. Common practice for marketing finisher barns is to utilize a ‘top-off’ strategy where a subset of the pen, generally the largest animals, are removed (or ‘pulled’) and marketed to provide increased pen space for the remaining pen-mates. Typically, there are multiple pulls with ranges from 2 to 5, depending on packer demand and pig growth rate. This strategy allows the opportunity to capture lost gain efficiency through compensatory growth in the remaining animals because of greater space availability. A caveat to the benefit of increased growth in the remaining pen mates is that near market weight pigs are beyond the linear phase of lean tissue growth, where the rate of lean tissue accretion declines and adipose tissue accretion increases [14]. This means that the ratio of fat:lean gain is increased. However, the data in this study suggest the opposite effect, with the lowest space allocation having the leanest body composition [15]. Further, many packers pay on a carcass lean basis rather than liveweight, due to consumer demands for a lean protein product. While the top-off strategy provides a practical solution to heavy market weight pigs, little is known about whether the composition of the liveweight gain following each pull limits the overall benefit of the top-off strategy. This also means the model, and hence recommendations, of Buhr et al. [8] may not be relevant for current production practices.

## 5. Conclusions

This study concluded that while the lowest pen space per pig (i.e., 0.61 m^2^) did not significantly impact overall growth performance by the final marketing event, it did tend to negatively affect BW as the pigs started to reach weights greater than current market weights (i.e., 127 kg). Additionally, in this study, the top-off strategy appears to have ameliorated the downside of slower initial growth, and the genetics of the pigs and management during this study allowed for lean tissue accretion to occur at a higher body weight, leading to higher carcass value. However, space allocation below the current recommendations may not be beneficial to production. Thus, careful consideration of space allowance and barn marketing strategy is important to ensure optimal growth and carcass quality in heavier-weight finishing pigs.

## Figures and Tables

**Table 1 animals-15-01451-t001:** Composition of diets (%, as-fed basis).

Ingredient, %	100 to 120 kg	120 to 140 kg
Corn	81.13	85.90
Soybean meal	16.25	11.50
Corn oil	1.00	1.00
Calcium	0.98	1.02
Salt	0.37	0.31
VTM	0.25	0.25
Copper chloride	0.03	0.03
Analyzed nutrients		
Crude protein, %	16.6	16.0
Ca, %	0.55	0.60
P, %	0.33	0.34

**Table 2 animals-15-01451-t002:** Weekly mean body weight ^1^ (kg) of finishing pigs from 108.3 ± 0.7 kg to heavy market weight (>135 kg) that were housed in pens of either barrows or gilts or had a mixed-sex population.

	Pen Composition				
BW, kg	Barrow	Gilt	Mixed	SEM	*p*-Value
Day 0	111.2 ^a^	106.6 ^b^	107.3 ^b^	1.1	0.01
Day 7	118.0 ^a^	113.2 ^b^	113.8 ^b^	1.0	<0.01
Day 14	125.0 ^a^	119.7 ^b^	120.4 ^b^	1.0	<0.01
Day 21	131.2 ^a^	126.5 ^b^	127.2 ^b^	1.0	<0.01
Day 28	132.9	132.4	133.0	1.1	0.90
Day 37 ^2^	137.8	137.7	138.9	1.0	0.63
Day 43	139.8	141.1	142.6	1.3	0.26
Day 49	-	143.4	144.3	1.0	0.13

^1^ Values were determined by dividing the pen weight by the number of individuals in each respective pen. ^2^ Represents a final weigh day before marketing. ^a,b^ Different superscripts in the same row indicate that groups were statistically different (*p* < 0.05).

**Table 3 animals-15-01451-t003:** Weekly mean body weight ^1^ (kg) of finishing pigs from 108.3 ± 0.7 kg to heavy market weight (>135 kg) that were housed in pens providing 0.61, 0.75, or 0.88 m^2^ per pig.

	Pen Space per Pig				
BW, kg	0.61 m^2^	0.75 m^2^	0.88 m^2^	SEM	*p*-Value
Day 0	108.5	107.9	108.6	1.3	0.92
Day 7	114.7	114.8	115.6	1.2	0.86
Day 14	121.0	121.7	122.4	1.2	0.73
Day 21	127.0	128.8	129.2	1.1	0.34
Day 28	131.9	133.7	132.7	1.1	0.48
Day 37 ^2^	136.6 ^y^	138.4 ^x^	139.5 ^x^	0.9	0.09
Day 43	139.3 ^y^	142.4 ^x^	142.5 ^x^	1.1	0.07
Day 49	142.1	145.2	144.3	1.0	0.13

^1^ Values were determined by dividing the pen weight by the number of individuals in each respective pen. ^2^ Represents a final weigh day before marketing. ^x,y^ Different superscripts in the same row indicate that groups tended to be different (0.05 ≤ *p* < 0.10).

**Table 4 animals-15-01451-t004:** Average daily gain (ADG, kg/d) of finishing pigs from 108.3 ± 0.7 kg to heavy market weight (>135 kg) that were housed in pens providing 0.61, 0.75, or 0.88 m^2^ per pig.

	Pen Space per Pig				
ADG, kg/d	0.61 m^2^	0.75 m^2^	0.88 m^2^	SEM	*p*-Value
0 to 7	0.93	0.98	1.00	0.03	0.13
7 to 14	0.92	0.95	0.96	0.02	0.41
14 to 21	0.88 ^b^	1.02 ^a^	1.00 ^a^	0.03	<0.01
21 to 28	0.83	0.78	0.77	0.04	0.65
28 to 37	0.63	0.75	0.80	0.07	0.17
37 to 43	0.92	0.98	0.86	0.05	0.24
43 to 49	0.63	0.76	0.90	0.10	0.17

Different superscripts in the same row indicate that groups were statistically different (*p* < 0.05).

**Table 5 animals-15-01451-t005:** Average daily feed intake (ADFI ^1^, kg/d) of finishing pigs from 108.3 ± 0.7 kg to heavy market weight (>135 kg) that were housed in pens providing 0.61, 0.75, or 0.88 m^2^ per pig.

	Pen Space per Pig				
ADFI, kg/d	0.61 m^2^	0.75 m^2^	0.88 m^2^	SEM	*p*-Value
0 to 7	2.7	2.7	2.7	0.1	0.85
7 to 14	3.3	3.3	3.4	0.1	0.66
14 to 21	3.1 ^b^	3.4 ^a^	3.2 ^a,b^	0.1	0.03
21 to 28	3.1	3.2	3.3	0.1	0.50
28 to 37	2.7 ^y^	2.9 ^x,y^	2.9 ^x^	0.1	0.07
37 to 43	3.7	3.7	3.5	0.1	0.26
43 to 49	3.3	3.4	3.3	0.1	0.78

^1^ Values were determined by measuring the distance from the top of the feed to the top of the feeder and applying the equation: FL = 0.3863 × X2 + 3.5509 × X + 22.849. ^a,b^ Different superscripts in the same row indicate that groups were statistically different (*p* < 0.05). ^x,y^ Different superscripts in the same row indicate that groups tended to be different (0.05 ≤ *p* < 0.10).

**Table 6 animals-15-01451-t006:** Gain:feed of finishing pigs from 108.3 ± 0.7 kg to heavy market weight (>135 kg) that were housed in pens providing 0.61, 0.75, or 0.88 m^2^ per pig.

	Pen Space per Pig				
Gain:Feed	0.61 m^2^	0.75 m^2^	0.88 m^2^	SEM	*p*-Value
0 to 7	0.33 ^y^	0.36 ^x,y^	0.38 ^x^	0.02	0.08
7 to 14	0.24 ^y^	0.29 ^x^	0.29 ^x^	0.02	0.07
14 to 21	0.28	0.30	0.30	0.01	0.34
21 to 28	0.25	0.26	0.25	0.02	0.90
28 to 37	0.22	0.25	0.27	0.02	0.19
37 to 43	0.25	0.27	0.24	0.02	0.30
43 to 49	0.19	0.22	0.28	0.03	0.12

^x,y^ Different superscripts in the same row indicate that groups tended to be different (0.05 ≤ *p* < 0.10).

**Table 7 animals-15-01451-t007:** Overall hot carcass weight (HCW), lean percent, and carcass value of finishing pigs from 108.3 ± 0.7 kg to heavy market weight (>135 kg) that were housed in pens providing 0.61, 0.75, or 0.88 m^2^ per pig.

	Pen Space per Pig				
	0.61 m^2^	0.75 m^2^	0.88 m^2^	SEM	*p*-Value
HCW ^1^, kg	110.2	110.9	111.1	0.2	0.24
Lean, %	56.8 ^x^	56.8 ^x^	56.5 ^y^	0.05	0.08
Value, USD/45 kg	60.52 ^a^	60.38 ^a^	59.71 ^b^	0.14	0.03

^1^ Hot carcass weight reported by the commercial abattoir. ^a,b^ Different superscripts in the same row indicate that groups were statistically different (*p* < 0.05). ^x,y^ Different superscripts in the same row indicate that groups tended to be different (0.05 ≤ *p* < 0.10).

## Data Availability

The original contributions presented in this study are included in the article. Further inquiries can be directed to the corresponding author(s).

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
