# Peer review of "Effect of Pen Space Allowances on Growth Performance of Finishing Pigsâ€"

_animals, 2025, doi:10.3390/ani15101451_

Round 1
Reviewer 1 Report
Comments and Suggestions for Authors
You show high SEM values for ADG and BW but not for ADF and G:F. Could you please discuss this point? Could you generally give more information to your statistical testing?
Have you tested the distribution of your data and selected suitable statistical tests accordingly?
The first sentence of your conclusion does not fit to your results, since all pigs reached the same final weight.
Author Response
Comments 1: You show high SEM values for ADG and BW but not for ADF and G:F. Could you please discuss this point? Could you generally give more information to your statistical testing?
Response 1: The high SEM of BW is reflective of the variation within the population of animals in the barn, especially according to sex as the trial was started in the finisher stage. The SEM values for ADG are dependent on BW because ADG is a calculated value. Therefore, the high SEM values from BW are reflected in the SEM of the ADG. Alternatively, the low SEM of ADFI suggests that feed intake was less variable across the population. Finally, the SEM of G:F is dependent on the SEM of ADG and the SEM of ADFI and the calculated SEM would be intermediate.
Comments 2: Have you tested the distribution of your data and selected suitable statistical tests accordingly?
Response 2: A normal distribution of the data was tested and an appropriate analysis of variance was applied.
Comments 3: The first sentence of your conclusion does not fit to your results, since all pigs reached the same final weight.
Response 3: The authors agree that while statistically similar final body weight was achieved by the end of the trial, the pigs housed with the lowest pen space per pig tended to weigh less on the preceding two weigh days. Added Lines 152-153 "Specifically, all pigs were weighed up to d37 before pigs were started to be shipped for marketing". A note has been added to Table 3 and the text that d37 represents a final weigh day before marketing.
Reviewer 2 Report
Comments and Suggestions for Authors
This article evaluates the growth performance of heavy finishing pigs in different space allowances. If the industry is going to heavier pigs at market then updated data should be provided to make sure they have adequate space during production to ensure good welfare and performance. This is needed information for the industry to adapt if they continue to grow heavier pigs they need to make sure they adapt housing requirements along the way.
Intro
It would benefit the paper to include a little background on how space allowance affects welfare. Your first sentence could be improved with even just saying maximize floor space utilization and welfare. The public cares about the welfare of the pigs in these spaces - that they have enough space - so it would benefit to show you're considering public demands/concerns as well and producer needs.
Methods
Need to discuss where you decided these space allowances, and use a reference (e.g. Ag Guide, PQA+). Even your highest treatment was lower than what you indicate in a reference that's needed at the current avg finisher weight. The lowest space allowance you used does not meet either of those guidelines, so a producer could not be a part of the PQA+ Certification is they housed at 0.61m2/pig; a certification that is highly encouraged. Did you have an exemption with your IACUC to house pigs at that space requirement?
You say 127kg is the current market weight so why are you marketing pigs at 135kg? Is there any industry references you could use to show it's trending to that weight?
Indicate in methods when and how long (assuming 49d based on Table 3 but that's the first time I see anything about time) they were allocated to these treatment groups. It reads as if they were assigned immediately prior to transport not that they were housed in them through the grower/finisher phase of production, or that they weren't in these groups the entire time but only after tops were transported out.
Results
3.1.1 if barrows were statistically greater than is that not a sex difference? Did you look for sex*trt differences or just sex and space allowance?
Discussion
2nd paragraph if the top-off strategy is how you were then determining space allowance this wasn't clear in methods. If so, in results then are the greater space allowances by BW then just smaller number of pigs/pen not that they were housed in that trt the entire time of that phase of production?
I read from the results even if measures are ameliorated either with top-off strategy or even just seeing the changes short-term at the beginning, that the lowest space allowance is not beneficial to production. The lowest space allowance is also below acceptable standard of care. If there are no differences in 0.75 and 0.88m2/pig then it's worth indicating that the current guidelines (0.74m2/pig) may still be acceptable,(t least from a performance standpoint.
Author Response
This article evaluates the growth performance of heavy finishing pigs in different space allowances. If the industry is going to heavier pigs at market then updated data should be provided to make sure they have adequate space during production to ensure good welfare and performance. This is needed information for the industry to adapt if they continue to grow heavier pigs they need to make sure they adapt housing requirements along the way.
Intro
Comments 1: It would benefit the paper to include a little background on how space allowance affects welfare. Your first sentence could be improved with even just saying maximize floor space utilization and welfare. The public cares about the welfare of the pigs in these spaces - that they have enough space - so it would benefit to show you're considering public demands/concerns as well and producer needs.
Response 1: The first sentence of the Introduction was modified to add “animal welfare as well as” and the Chidgey, 2024 reference was moved forward. Also, the second sentence was modified: “Animal performance and welfare (e.g., pen hygiene, maintenance of health, and ability to express natural behaviors) generally declines as the space allocation per pig goes below a lower critical value”.
Methods
Comments 2: Need to discuss where you decided these space allowances, and use a reference (e.g. Ag Guide, PQA+). Even your highest treatment was lower than what you indicate in a reference that's needed at the current avg finisher weight. The lowest space allowance you used does not meet either of those guidelines, so a producer could not be a part of the PQA+ Certification is they housed at 0.61m2/pig; a certification that is highly encouraged. Did you have an exemption with your IACUC to house pigs at that space requirement?
Response 2: The pen space allowances were decided to be reflective of industry standards and available literature. For example, for pigs 110-130 kg, the minimum recommended space (based on the lower critical value k = 0.0336 (Gonyou et al., 2006) would be 0.78-0.88 m2. The Ag Guide recommendations are greater than industry standards, so were not used as a reference. PQA Plus does not prescribe exact pen space requirements but emphasizes that pigs must have enough space to lie down, stand up, and move without injury and thus evaluation of this parameter is subjective and at the discretion of the PQA auditor. The IACUC reviewed and approved all procedures.
Comments 3: You say 127kg is the current market weight so why are you marketing pigs at 135kg? Is there any industry references you could use to show it's trending to that weight?
Response 3: The researchers relied on statements within Johnston et al. (2017) and personal communications. The USDA Weekly Average Weight of Barrows and Gilts for week ending April 20, 2025 and for week ending April 21, 2024 reported the national average of 130 kg.
Comments 4: Indicate in methods when and how long (assuming 49d based on Table 3 but that's the first time I see anything about time) they were allocated to these treatment groups. It reads as if they were assigned immediately prior to transport not that they were housed in them through the grower/finisher phase of production, or that they weren't in these groups the entire time but only after tops were transported out.
Response 4: Lines 95-105 were modified: “Approximately one month before the start of the pen space study described in this report, pigs from the barn were reallocated to create 15 barrow, 15 gilt, and 15 mixed sex pens, with the mixed sex pens housing approximately equal numbers of gilts and barrows. Barrow, gilt and mixed pens were randomly assigned within the facility. At the start of and for the duration of the 49 d trial, pen space was adjusted by changing the position of the front gates. Pens with the front gate adjusted as far back as the feeder provided 0.61 m2/pig, while pens with the front gate placed equidistant between the standard position and the furthest back position provided 0.75 m2/pig; the front gate of the pen in standard position provided pen space of 0.88 m2/pig. Each space allocation was replicated 5 times within each sex, resulting in a total of 15 pens per space allocation”.
Results
Comments 5: 3.1.1 if barrows were statistically greater than is that not a sex difference? Did you look for sex*trt differences or just sex and space allowance?
Response 5: The majority of the first pigs removed from the barn in the first marketing event were barrows. Thus, sex difference was not observed in the rest of the experiment. The analysis was limited to sex and space allowance. Lines 150-155 have been modified.
Discussion
Comments 6: 2nd paragraph if the top-off strategy is how you were then determining space allowance this wasn't clear in methods. If so, in results then are the greater space allowances by BW then just smaller number of pigs/pen not that they were housed in that trt the entire time of that phase of production?
Response 6: The pen space allowances were decided and maintained through the trial to be reflective of industry standards and available literature. For example, for pigs 110-130 kg, the minimum recommended space (based on the lower critical value k = 0.0336 (Gonyou et al., 2006) would be 0.78-0.88 m2. The pens were stocked with an appropriate and equal number of pigs such that the standard position provided 0.88 m2 per pig, the position of the front gate equidistant between the standard position and the furthest back position provided 075 m2, and the furthest back position provided 0.61 m2.
Comments 7: I read from the results even if measures are ameliorated either with top-off strategy or even just seeing the changes short-term at the beginning, that the lowest space allowance is not beneficial to production. The lowest space allowance is also below acceptable standard of care. If there are no differences in 0.75 and 0.88m2/pig then it's worth indicating that the current guidelines (0.74m2/pig) may still be acceptable,(t least from a performance standpoint.
Response 7: The authors agree with that observation. Lines 278-279 added: “However, space allocation below current recommendations may not be beneficial to production”.
Reviewer 3 Report
Comments and Suggestions for Authors
The authors set out to investigate the impact of floor space allocation on growth performance of finishing pigs. The topic is interesting and could provide information to both producers and other stakeholders in the swine industry about potential production and management strategies to increase profitability or mitigate unnecessary production losses. The manuscript is concise and overall well written but I have some concerns about the clarity of the material and methods and study setup as well as the statistical method used.
Line by line comments are provided below:
Abstract:
Ln 14-17: Consider revising this sentence to give the reader the context of what is consider a ‘lighter’ market weight pig. You write that you want to investigate pigs from 108.3 ± 0.7 kg. This seems very specific for a pre-study hypothesis/objective, please clarify this specific weight in the material and methods as your cut-off as it is currently not explained elsewhere in the manuscript. The objective at the end of the introduction also specifies investigating pigs >135kg.
Ln 21: Here and elsewhere, double check that your p-values are written out consistently. (P<) vs (p <).
Material and methods:
Ln 90-91: Just to clarify; the three allocated pen space categories were based on pen configuration restrictions, but adjusted to still incrementally (+0.14 and +0.13 m2) increase and fall within the range of the 0.67 -0.90 m2/pig from the literature? Is that correctly interpreted?
Ln 96-97: For replication purposes, could you please provide the name/model and manufacturer of the scale?
Ln 116-117. Please double check the removal days.
Statistical evaluation:
I am not sure I am quite understanding the trial logistics and the statistical approach taken. You have 15 pens with 23 pigs per sex/mixed allocation (barrow/mixed sex/gilts) (n=15). When are they transferred into their separate treatment groups? Are 5 pens with 23 pens allocated for each (barrow/mixed/gilts) per pen space treatment? Are new animals entering the groups as others are leaving or are the groups getting smaller over time? It seems that they are not stable by either individual nor number of pigs?
If group dynamics are changing, then at least part of the study setup is pseudoreplicated (HCW) as the individual pig can’t be considered independent, as the carcass weight would have been influenced by the group dynamics over time, social interactions, space allocation, resource access etc. (This will also be true for pen means as pen dynamics would influence the outcome). This needs to be clarified further in the manuscript and remodeled based on what pen and treatment that pig originated from. Your N-value for the HCW analysis will still be 15 per floor allocation treatment and not 330, 282, and 332.
You are stating that you are not attributing a time variable to the models, but you are taking 8 measurements over time for the same pens. Would any pig from a previous measurement remain in the same pen 7 days later? Did you perform individual models for each time point? What was the rationale?
If you are doing multiple observations over time for pens that would include the same or some of the original pigs that would be a repeated measures ANOVA with pig-ID nested within pen as a random effect and possibly a post-hoc Bonferroni correction or similar for pair-wise comparisons.
I would also provide F-statistics for the models used.
Results:
Ln 136-139. Unclear if you are referring to previously published results only, or if you investigated sex as an independent variable? Regardless, please move the citation to the discussion section and if investigated, add this information to the statistics section in the material and methods and provide at minimum the BW +/- SEM in the result text (or table if you prefer) for each of the barrow, mixed-sex and gilt groups up to marketing began.
Table 3. Please add legend to provide explanation for X-Y superscripts.
Table 7. Please adjust superscripts to be consistent with both legend and tendency superscripts provided elsewhere.
Author Response
The authors set out to investigate the impact of floor space allocation on growth performance of finishing pigs. The topic is interesting and could provide information to both producers and other stakeholders in the swine industry about potential production and management strategies to increase profitability or mitigate unnecessary production losses. The manuscript is concise and overall well written but I have some concerns about the clarity of the material and methods and study setup as well as the statistical method used.
Line by line comments are provided below:
Abstract:
Comments 1: Ln 14-17: Consider revising this sentence to give the reader the context of what is consider a ‘lighter’ market weight pig. You write that you want to investigate pigs from 108.3 ± 0.7 kg. This seems very specific for a pre-study hypothesis/objective, please clarify this specific weight in the material and methods as your cut-off as it is currently not explained elsewhere in the manuscript. The objective at the end of the introduction also specifies investigating pigs >135kg.
Response 1: Line 16: added “than currently in the United States (i.e. ~130 kg)”
Line 18: replaced “108.3 ± 0.7 kg” with “over 49 d” because the starting weight was a function of start date of the trial, rather than a specific weight to initiate the trial. Also, “(n=1,035; initial body weight (BW) 108 ± 0.7 kg)” was added to Line 72.
Comments 2: Ln 21: Here and elsewhere, double check that your p-values are written out consistently. (P<) vs (p <).
Response 2: This has been checked.
Material and methods:
Comments 3: Ln 90-91: Just to clarify; the three allocated pen space categories were based on pen configuration restrictions, but adjusted to still incrementally (+0.14 and +0.13 m2) increase and fall within the range of the 0.67 -0.90 m2/pig from the literature? Is that correctly interpreted?
Response 3: The reviewer is correct.
Comments 4: Ln 96-97: For replication purposes, could you please provide the name/model and manufacturer of the scale?
Response 4: The scale information has been added to the manuscript (Line 108).
Comments 5: Ln 116-117. Please double check the removal days.
Response 5: This has been checked.
Statistical evaluation:
Comments 6: I am not sure I am quite understanding the trial logistics and the statistical approach taken. You have 15 pens with 23 pigs per sex/mixed allocation (barrow/mixed sex/gilts) (n=15). When are they transferred into their separate treatment groups? Are 5 pens with 23 pens allocated for each (barrow/mixed/gilts) per pen space treatment? Are new animals entering the groups as others are leaving or are the groups getting smaller over time? It seems that they are not stable by either individual nor number of pigs?
Response 6: Lines 95-105 were modified: “Approximately one month before the start of the pen space study described in this report, pigs from the barn were reallocated to create 15 barrow, 15 gilt, and 15 mixed sex pens, with the mixed sex pens housing approximately equal numbers of gilts and barrows. Barrow, gilt and mixed pens were randomly assigned within the facility. At the start of and for the duration of the 49 d trial, pen space was adjusted by changing the position of the front gates. Pens with the front gate adjusted as far back as the feeder provided 0.61 m2/pig, while pens with the front gate placed equidistant between the standard position and the furthest back position provided 0.75 m2/pig; the front gate of the pen in standard position provided pen space of 0.88 m2/pig. Each space allocation was replicated 5 times within each sex, resulting in a total of 15 pens per space allocation". Animals were not moved between pens during the study.
Comments 7: If group dynamics are changing, then at least part of the study setup is pseudoreplicated (HCW) as the individual pig can’t be considered independent, as the carcass weight would have been influenced by the group dynamics over time, social interactions, space allocation, resource access etc. (This will also be true for pen means as pen dynamics would influence the outcome). This needs to be clarified further in the manuscript and remodeled based on what pen and treatment that pig originated from. Your N-value for the HCW analysis will still be 15 per floor allocation treatment and not 330, 282, and 332.
Response 7: The authors agree that while the experimental unit for growth performance is pen and thus one might also consider pen the best experimental unit for HCW, it is not feasible to retain pen units on the truck used to move pigs to market. Pigs can however, be tattooed by treatment and tracking of individual pigs at the plant allows individual HCW by treatment. Thus it is typical to consider the HCW of individual animals even though they originated from pens that would be otherwise the experimental unit.
Comments 8: You are stating that you are not attributing a time variable to the models, but you are taking 8 measurements over time for the same pens. Would any pig from a previous measurement remain in the same pen 7 days later? Did you perform individual models for each time point? What was the rationale?
Response 8: No pigs moved between pens after the start of the experiment. The authors contend that the same ANOVA results are obtained by considering ‘pen’ as a random variable rather than as a repeated variable.
Comments 9: If you are doing multiple observations over time for pens that would include the same or some of the original pigs that would be a repeated measures ANOVA with pig-ID nested within pen as a random effect and possibly a post-hoc Bonferroni correction or similar for pair-wise comparisons.
Response 9: The authors contend that the same ANOVA results are obtained by considering ‘pen’ as a random variable rather than as a repeated variable.
Comments 10: I would also provide F-statistics for the models used.
Response 10: Our model includes random effects (i.e. pen) making F-statistics less straightforward to interpret, especially with the use of Kenward-Roger degrees of freedom method.
Results:
Comments 11: Ln 136-139. Unclear if you are referring to previously published results only, or if you investigated sex as an independent variable? Regardless, please move the citation to the discussion section and if investigated, add this information to the statistics section in the material and methods and provide at minimum the BW +/- SEM in the result text (or table if you prefer) for each of the barrow, mixed-sex and gilt groups up to marketing began.
Response 11: An additional table has been added to the Results section. The information has also been added to the Statistical evaluation section. The citation has been moved to the discussion (Lines 234-235): "Similarly, barrow only pens weighed more from the beginning of the trial up to the first marketing, as previously reported [13]".
Comments 12: Table 3. Please add legend to provide explanation for X-Y superscripts.
Response 12: The additional footnote has been added.
Comments 13: Table 7. Please adjust superscripts to be consistent with both legend and tendency superscripts provided elsewhere.
Response 13: The ‘Lean, %’ superscripts were corrected and the additional footnote has been added.
Round 2
Reviewer 1 Report
Comments and Suggestions for Authors
Table3: which pen space?
Table4: Pen space per pig. this values is not constant, if pigs are removed. If pigs are removed, this should be indicated in the table.
n gilt, n barrow?
weekly mean body weight is not described in material and methods
again: Could you please describe the statistics in more detail. Which kind of distribution test was used? Otherwise it will be impossible to reproduce the results by other scientists. It is not sufficient to name the software.
Author Response
Comments 1: Table3: which pen space?
Response 1: The weekly mean body weights presented in Table 3 were not stratified by pen space. Instead, the body weights in the table represent the average of all the pens of each sex, as identified by the table column headers.
Comments 2: Table4: Pen space per pig. this values is not constant, if pigs are removed. If pigs are removed, this should be indicated in the table.
Response 2: The authors recognize that the pen space per pig changed as pigs were removed for marketing. However, the originally applied pen space per pig remains as the treatment effect being investigated. Furthermore, there were no differences observed after the first pigs were removed for marketing. Also, a footnote has been added to Table 4 “2 Represents a final weigh day before marketing.”
Comments 3: n gilt, n barrow?
Response 3: The authors recognize that sex of the animals remaining after marketing could influence the population dynamics of each pen. However, the sex of the removed animals was not recorded. As noted above, the lack of difference in performance after the first market removal suggests that sex of animals remaining had minor effect in this case; however, this could be included in any future studies.
Comments 4: weekly mean body weight is not described in material and methods
Response 4: The weekly mean body weight has been described in Lines 107-111.
Comments 5: again: Could you please describe the statistics in more detail. Which kind of distribution test was used? Otherwise it will be impossible to reproduce the results by other scientists. It is not sufficient to name the software.
Response 5: “using the UNIVARIATE procedure” was added in Lines 135-136
“allowing for both fixed and random effects fitting a mixed linear model” was added in Lines 136-137
Reviewer 2 Report
Comments and Suggestions for Authors
The information added has helped make the objective and design more clear.
Author Response
Comments 1: The information added has helped make the objective and design more clear.
Response 1: The authors thank the reviewer for their feedback and comments that have improved the manuscript.
Reviewer 3 Report
Comments and Suggestions for Authors
Dear authors,
Thank you for the updated version of the manuscript. I think the manuscript has been greatly improved and responses given provided additional clarification. I believe that you have satisfactorily responded to all of my questions/suggestions in the previous review.
That said, we might still be slightly in disagreement on the optimal statistical method use and/or description for parts of the study (which is fine). I still believe a higher level of detail could have been incorporated and provided by accumulating data into a repeated measures model for gain:feed, feed intake and ADG using day as your time variable and with the calculated individual variation over time. However, as you claim that no changes in outcome was observed between the models/approach, I am not going to challenge this further and let the reader have the final interpretation.
Thank you so much for the efforts and good luck!
Author Response
Comments 1: Dear authors,
Thank you for the updated version of the manuscript. I think the manuscript has been greatly improved and responses given provided additional clarification. I believe that you have satisfactorily responded to all of my questions/suggestions in the previous review.
That said, we might still be slightly in disagreement on the optimal statistical method use and/or description for parts of the study (which is fine). I still believe a higher level of detail could have been incorporated and provided by accumulating data into a repeated measures model for gain:feed, feed intake and ADG using day as your time variable and with the calculated individual variation over time. However, as you claim that no changes in outcome was observed between the models/approach, I am not going to challenge this further and let the reader have the final interpretation.
Thank you so much for the efforts and good luck!
Response 1: The authors thank the reviewer for their feedback and comments that have improved the manuscript.